# Prolongation of Acid-Fast Bacilli Sputum Smear Positivity in Patients with Multidrug-Resistant Pulmonary Tuberculosis

**DOI:** 10.3390/pathogens12091133

**Published:** 2023-09-05

**Authors:** Sidwell Mvo, Carine Bokop, Benjamin Longo-Mbenza, Sandeep D. Vasaikar, Teke Apalata

**Affiliations:** 1Division of Medical Microbiology, Department of Pathology & Laboratory Medicine, Faculty of Health Sciences, Walter Sisulu University, Mthatha 5100, South Africa; smvo@wsu.ac.za (S.M.); carine.bokopf@gmail.com (C.B.); longombenza@gmail.com (B.L.-M.); 2Vaccine and Infectious Disease Analytics Research Unit, University of the Witwatersrand, Johannesburg 2000, South Africa; 3Department of Medical Microbiology, National Health Laboratory Services, Nelson Mandela Academic Complex, Mthatha 5100, South Africa; sandeepvasaikar@yahoo.com

**Keywords:** MDR-TB, HIV status, time to sputum microscopy conversion, smear positivity

## Abstract

The study sought to determine factors associated with prolonged smear positivity in multidrug-resistant tuberculosis (MDR-TB) patients following appropriate management. Newly diagnosed patients were enrolled between June 2017 and May 2018. Sputum samples were collected for Xpert^®^ MTB/RIF and line probe assays (LiPAs). Microscopic tests were performed at baseline and 4, 8, and 12 weeks post-anti-TB therapy. Of the 200 patients, 114 (57%) were HIV-positive. After 12 weeks of treatment, there was a significant microscopy conversion rate among DS-TB patients compared to MDR-TB patients irrespective of their HIV status (*p* = 0.0013). All MDR-TB patients who had a baseline smear grade ranging from scanty to +1 converted negative, while 25% ranging from +2 to +3 remained positive until the end of 12 weeks (*p* = 0.014). Factors associated with smear positivity included age <35 years (*p* = 0.021), initial CD4+ T-cell count ≥200 cells/mm^3^ (*p* = 0.010), and baseline smear grade ≥2+ (*p* = 0.014). Cox regression showed that only the baseline smear grade ≥2+ was independently associated with prolonged smear positivity in MDR-TB patients (*p* = 0.011) after adjusting for HIV status, CD4+ T-cell count, and age. Baseline sputum smear grade ≥2+ is a key determinant for prolonged smear positivity beyond 12 weeks of effective anti-TB therapy in MDR-TB patients.

## 1. Introduction

Ranked only behind India and Russia, South Africa occupies the third position because the country has a high number of patients diagnosed with drug-resistant tuberculosis (DR-TB) in the world [1]. The rise of DR-TB cases in South Africa and other parts of the world threatens the efforts to end TB [1]. Globally, studies have shown that one of the major public health challenges today is the fact that numerous countries, particularly countries with limited resources, have high rates of multidrug-resistant tuberculosis (MDR-TB) and extensive drug-resistant tuberculosis (XDR-TB) [2,3]. It is evident through various published reports that most of those MDR-TB and XDR-TB cases are driven by the HIV epidemic [2,3]. In addition, the successful response to anti-TB treatment has been also challenged. For example, in 2014, reports underlined the fact that more than half of the total estimated cases of DR-TB were receiving anti-TB therapy and only half achieved treatment success [2,3].

Among strategies able to reduce the transmission of tuberculosis (including MDR-TB and XDR-TB), early detection of cases and their effective management are proven to be the cornerstone approach for substantially preventing the occurrence of new TB cases [4,5]. In patients diagnosed with TB, the conversion of smear microscopy is achieved when the initial positive result becomes negative following an appropriate anti-TB therapy [6]. In order to achieve such a conversion of smear microscopy during the first two months of the intensive care phase, patient compliance with anti-TB drugs is mandatory, hence representing one of the most successful outcomes of any TB control program [7,8]. In settings with limited resources, smear conversion is, therefore, an important indicator of patient response to therapy because the use of culture techniques for monitoring patient response is costly [4]. In addition to patient adherence to TB treatment, studies have shown that demographic factors (age and gender), HIV status, and initial level of infectiousness (measured by smear grade) are determinants for sputum conversion rate. HIV-infected patients have a 5–15% probability of developing active TB disease, making HIV coinfection one of the most significant risk factors for developing TB [2,3]. It has been indicated that HIV prevalence in a setting is proportionally correlated with TB incidence. About 8% of all new TB cases can be attributed to HIV [9]. In 2018, in South Africa, HIV prevalence was estimated to be 13% at the population level, while 6 out of 10 TB patients were found to be coinfected with HIV [10]. Moreover, the probability of death was two times more likely to occur among TB and HIV coinfected patients compared to HIV-infected patients without TB, even after adjusting for the levels of CD4+ T-cells and the initiation of antiretroviral therapy [11].

In South Africa, the Xpert MTB/RIF (Cepheid, Sunnyvale, USA) has largely replaced direct smear microscopy for the routine detection of *Mycobacterium tuberculosis* even at primary care levels. However, the microscopy smear test remains useful in patients who are confirmed to have tuberculosis for monitoring changes in loads of *M. tuberculosis* observed during microscopy testing using smear grading, hence indicating treatment response in patients whose initial smear-positivity results were determined [7,8]. Although finding acid-fast bacilli (AFB) is highly specific in confirming the diagnosis of smear-positive tuberculosis, the viability of AFB seen can only be confirmed by cultures. Nevertheless, in many developing countries, microscopy conversion (from positive to negative results) is still considered a predictor of survival and quality care because TB transmission is highly likely to occur when a TB patient has positive sputum microscopy. However, information from the literature pertaining to the time to sputum microscopy conversion among MDR-TB patients coinfected with HIV is scanty with enormous controversies. This study determined the impacts of HIV disease progression, baseline TB infectiousness rate, and demographic factors (age and gender) on rates and time to sputum microscopy conversion among MDR-TB patients following the initiation of an appropriate anti-TB therapy. It was intended that in resource-limited settings, the use of baseline microscopy smear grading might also play a key role in risk stratification before treatment initiation of MDR-TB patients with respect to TB transmission and rational use of basic IPC supplies (i.e., surgical and N95 masks).

## 2. Materials and Methods

### 2.1. Study Setting, Design, and Population

This was a prospective cohort study. Patients attending health facilities in OR Tambo district whose sputum samples were received in the TB laboratory at the National Health Laboratory Services (NHLS) in Nelson Mandela Academic Hospital were enrolled. A presumptive diagnosis of TB was made using clinical and radiological information. Data on patient demographics (age and gender), HIV status, and baseline sputum microscopy smear grading were recorded. The final inclusion criteria for the study included all newly diagnosed TB patients between June 2017 and May 2018, aged ≥18 years, and provided written consent to participate in the study. A convenient sample of 200 newly diagnosed patients was selected, of which 100 were MDR-TB and the remaining 100 were DS-TB patients. These patients were further grouped according to their HIV status. For HIV-positive patients, levels of immunosuppression were measured and classified as severe (CD4 < 200 cells/mm^3^), advanced (CD4 = 200–349 cells/mm^3^), and mild (CD4 = 350–499 cells/mm^3^) [12]. Patients aged <18 years and those who refused to provide consent were excluded from the study. All TB-diagnosed patients with infection other than HIV/AIDS were also excluded from the study.

### 2.2. Microbiological Testing

Sputum samples (expectorated or induced) were collected from all new patients suspected of having TB. Specimens were transported into the NHLS TB laboratory within 4 h of collection and were processed for GeneXpert MTB/RIF, Line Probe Assays (LiPAs), and microscopy (Fluorescence microscopy and Ziehl Neelsen). Before the process, macroscopic characteristics of sputum were determined as purulent, mucoid, saliva, or blood-stained. Blood samples were collected for HIV testing and CD4+ T-cell counts.

### 2.3. Microscopic Staining Techniques

Direct smears were prepared in duplicate, from sputum sediment after decontamination (by the sodium Hydroxide-N-Acetyl-L-Cysteine method), for staining by Ziehl Neelsen (Zn) carbol fuchsin and Fluorescent Auramine O staining techniques [13]. Staining using ZN was carried out according to standard published procedures, and slides were examined with bright-field microscopy (Olympus CX21) using the 100× oil objective (10× eyepiece for a total of 1000× magnification). For staining with Auramine O, a fluorescent microscope was used, and microscopists scanned the entire smear with the 20× objective (with 10× eyepiece for a total of 200× magnification). Using 20× magnification, one 2 cm length is equivalent to 30 fields, which is sufficient to report a negative result. The presence or absence of AFB was reported using WHO/IUATLD guidelines [13]. All smears were read by two microscopists involved in routine diagnostic services; in case of discrepant results, a third microscopist reviewed the slides. Positive (*M. tuberculosis* H37Rv strain) and negative (sterile distilled water) control slides were included with every batch of patient slides and when new lots of stains were received.

Samples were initially examined using the Fluorescent Auramine O staining technique. All positive results were thereafter confirmed using Ziehl Neelsen (Zn) carbol fuchsin technique. Microscopy smear results were used to estimate the levels of infectiousness of each patient at the initiation of effective treatment (enrolment) and also during follow-ups at 4, 8, and 12 weeks.

### 2.4. Molecular Methods

The GeneXpert MTB/RIF assay (Cepheid, Sunnyvale, CA, USA) was performed directly on TB samples using version G4 of cartridges according to the manufacturer’s recommendations. Two milliliters of the final samples were transferred into the Xpert MTB/RIF cartridge, and the latter was loaded into the GeneXpert instrument. Reported results were *M. tuberculosis*-negative or -positive and Rifampicin (RIF)-resistant (presence of mutations in Rifampicin Resistance-Determining Region of *M. tuberculosis*) or -susceptible [14].

Line probe assays (LiPAs: Hain Lifescience, Nehren, Germany) were used to simultaneously diagnose TB and detect mutations conferring Rifampicin and Isoniazid resistance by detecting the presence of the DNA of *M. tuberculosis* in the sputum and also identifying any mutations in the DNA that may cause rifampicin and/or isoniazid resistance. The test included the simultaneous detection of the most important *rpoB* mutations, which confer Rifampicin resistance, and *katG* and *inhA* mutations, which confer high-level and low-level Isoniazid (INH) resistance, respectively [14].

### 2.5. HIV Testing and Definitions of HIV-Induced Immunosuppression

An initial HIV test was performed on blood using the HIV rapid test Determine HIV-1/2/O (Abbott Laboratories, Abbott Park, IL, USA). Patients with negative HIV results were retested 3 months later. All initial HIV-positive samples were subsequently retested using a second HIV screening kit (ARCHITECT HIV Ag/Ab Combo Reagent Kits, ABBOTT, Wiesbaden, Germany) and HIV confirmatory test kit using Eleyses HIV Combi kit (Abbot Architect Plus).

As part of the routine management of the patients, all HIV-infected patients benefited directly from CD4+ T-cell count measurements and HIV viral loads. Absolute values of CD4+ T-cell counts (cells/mm^3^) were used to determine the degree or severity of immunocompromise following the World Health Organization’s (WHO) immunological staging criteria: CD4 levels < 200/mm^3^ (severe immunosuppression), CD4 levels 200–349/mm^3^ (advanced immunosuppression), and CD4 levels 350–499/mm^3^ (mild immunosuppression) [12,15].

### 2.6. Treatment Regimens of DR-TB Patients

Patients diagnosed as MDR-TB or RIF-resistant TB, with or without additional resistance were initiated on the standardized treatment given in two phases (intensive and continuation phases) [14]. The total duration of treatment for the MDR-TB regimen was a function of either a short or long regimen. Each patient was treated with an individualized treatment regimen composed of second-line injectable drugs (amikacin or capreomycin), fluoroquinolones (levofloxacin, ofloxacin, or moxifloxacin), ethambutol, pyrazinamide, and high-dose isoniazid. The short regimen included 4–6 months of kanamycin (Km)–moxifloxacin (Mfx)–prothionamide (Pto)–clofazimine (Cfz)–pyrazinamide (Z)–high-dose isoniazid (H)–ethambutol (E) followed by 5 months of Mfx–Cfz–Z–E, while the long regimen included 6 months of Z–Km(Am)–Mfx–Eto–terizidone (Trd) followed by 18 months of Z–Mfx–Eto–Trd [14]. Patients were reviewed at monthly intervals during the intensive phase and at 3-month intervals during the continuation phase until the end of treatment.

### 2.7. Data Variables and Statistical Analysis

Statistical Product and Service Solutions (SPSS) version 23 was used for all statistical analyses. The level of significance was set at ≤0.05. Categorical variables were expressed as proportions (%), and continuous (discrete ordinal) variables were presented as mean (±standard deviation). For bivariate analyses, the Chi-square (*X*^2^) test was used to compare proportions, and T-test for independent samples was used to compare two groups, and the analysis of variance (ANOVA) was used to compare more than two groups. Trend Chi-square test was used when the proportion of cells with expected frequency under 5 was found to be more than 20%. Kaplan–Meier analysis curves (survival curves) were constructed for microscopy conversion of HIV-positive and HIV-negative patients with MDR-TB receiving treatment, and the Wilcoxon rank sum test was used to compare two or more survival curves. Finally, multivariate Cox regression analysis was used to identify independent determinants associated with sputum microscopy smear conversion.

### 2.8. Ethics Approval and Consent to Participate

Ethical clearance was obtained from the Research Ethics and Biosafety Committee of the Faculty of Medicine at Walter Sisulu University (Ethics Ref. No. 107/2014), and permission to conduct the study was obtained from the hospital management at Nelson Mandela Academic Hospital in Mthatha complex and from the National Health Laboratory Services (NHLS). Patients were enrolled following a written informed consent, and in order to protect the privacy and confidentiality of patients, no names were recorded. Instead, a personalized research number was used for each patient, and only investigators had access to the collected data.

## 3. Results

A total of 200 patients were enrolled in this study, of which 103 (51.5%) were males and 97 (48.5%) were females, with a sex ratio of males to females of 1.06:1. Out of these 200 patients, 55 (27.5%) were aged ≥35 years, while 145 (72.5%) were aged <35 years. The mean age of the study population was 34.8 (±11.6) years ranging from 18 to 67 years. HIV test results were obtained from the 200 participants, of which 114 (57%) tested HIV-positive and 86 (43%) tested HIV-negative (Table 1). Of the 200 participants, 100 (50%) were confirmed as having multidrug-resistant tuberculosis (MDR-TB), while the remaining 100 (50%) patients were diagnosed with drug-susceptible tuberculosis (DS-TB).

### 3.1. Sputum Microscopy Smear Conversion Rates and Time to Sputum Conversion among HIV-Infected and -Uninfected Patients

Table 2 below depicts the number of cases of microscopy smear grades at baseline stratified by HIV serostatus of the participants. HIV-infected and -uninfected patients were matched by their microscopy smear grades at enrolment (*p* = 0.794).

After 4 and 8 weeks of appropriate anti-TB treatment, there was no statistically significant difference in the microscopy smear conversion rates of HIV-infected patients as compared to HIV-uninfected patients. However, Table 3 below shows that after 12 weeks of appropriate anti-TB treatment, there was a significant microscopy conversion rate among DS-TB patients as compared to MDR-TB patients, irrespective of their HIV status (*p* = 0.0013).

Using the Wilcoxon rank sum test, there was no significant difference in time to microscopy conversion when comparing survival curves of HIV-positive patients aged ≥35 years (5.96 ± 0.387 weeks), HIV-positive patients below 35 years (6.78 ± 0.40 weeks), HIV-negative patients aged ≥35 years (5.83 ± 0.47 weeks), and HIV-negative patients below 35 years (7.06 ± 0.48 weeks) (Chi-square = 4.981; *p* = 0.173).

Of the 200 patients, 114 (57%) were HIV-infected of which we only obtained CD_4_+ T-cell count results from 50 patients. The mean CD_4_+ T-cells for the 50 patients was 179.84 cells/mm^3^, ranging from 3 to 487 cells/mm^3^, with a standard error of the mean (SEM) of 19.59 cells/mm^3^ and standard deviation (SD) of 139.9 cells/mm^3^. Of the 50 HIV-infected patients, 28 (56%) had a CD_4_+ T-cell count ≤ 200 cells/mm^3^ (severe immune suppression), while 17/50 (34%) and 5/50 (10%) HIV-infected patients had a CD_4_+ T-cell count of 200–349 cells/mm^3^ (advanced immune suppression) and 350–499 cells/mm^3^ (mild immune suppression), respectively. A linear trend between levels of immune suppression by means of CD4 T-cell count (cell/mm^3^) and sputum smear conversion status among MDR-TB patients was determined, and the findings show that the value of the Chi-squared test for Trend statistic was 1.356 and the *p*-value was 0.244 as depicted in Table 4. These results show that there is no significant upward trend in the proportion of MDR-TB patients who converted negative after 12 weeks of appropriate anti-TB therapy classified by immune suppression levels on treatment initiation.

However, there was a significant difference between the above levels of immuno-suppression (CD4+ T-cell count at the beginning of anti-TB therapy) and time to microscopy smear conversion. The latter was shorter (4.25 ± 2.43 weeks) among HIV-positive patients whose CD4+ T-cell counts were <200 cells/mm^3^ as compared to patients who had CD4+ T-cell counts between 200 and 349 cells/mm^3^ (6.77 ± 3.99 weeks) and 350 and 499 cells/mm^3^ (7.40 ± 2.97 weeks) (ANOVA; *p* = 0.015).

Furthermore, when the above 50 HIV-positive patients coinfected with MDR TB were categorized into two groups composed of Group 1: 28 (56%) patients with CD4+ T-cell counts <200 cells/mm^3^ and Group 2: 22 (44%) patients with CD4+ T-cell counts ≥200 cells/mm^3^, patients with CD4+ T-cell counts <200 cells/mm^3^ converted faster (5.379 ± 0.456 weeks; 95% CI: 4.486–6.273) than those with CD4+ T-cell counts ≥200 cells/mm^3^ (8.000 ± 0.746 weeks; 95% CI: 6.538–9.462), and this difference reached statistical significance using the Wilcoxon rank sum test (*p* = 0.004) as depicted in Figure 1 below.

### 3.2. Factors Associated with Prolongation of Sputum Smear Positivity in All TB Patients at the End of 12 Weeks Post-Anti-TB Initiation

Using bivariate analysis, Table 5 shows that at the end of 12 weeks of appropriate anti-TB therapy, patients aged <35 years (*p* = 0.045) who had a baseline sputum smear grade from 2+ to 3+ on microscopy (*p* < 0.0001) and were diagnosed with MDR-TB (*p* = 0.003) were significantly associated with a prolongation of their sputum smear positivity.

At the end of 12 weeks of anti-TB therapy, 9% of all DS-TB patients remained positive versus 25% of all their MDR-TB counterparts (*p* = 0.003). Figure 2 shows that time to sputum smear conversion of DS-TB patients was shorter (5.92 ± 0.26 weeks; CI: 5.403–6.437) than MDR-TB patients (7.00 ± 0.34 weeks; CI: 6.328–7.672) using the Wilcoxon rank sum test (Chi-square = 3.921; *p* = 0.048).

In addition, 87.9% of MDR-TB patients who were aged ≥35 years converted negative at the end of 12 weeks of anti-TB therapy as compared to 78.2% of MDR-TB patients aged <35 years (*p* = 0.045). Figure 3 shows that time to sputum smear conversion for patients aged ≥35 years was shorter (5.91 ± 0.29 weeks; CI: 5.327–6.495) than the time to conversion for patients aged below 35 years (6.91 ± 0.31 weeks; CI: 6.302–7.516) (Chi-square = 4.875; *p* = 0.027).

As displayed in Table 6 and Figure 4 below, there was a significant difference between baseline sputum smear grades and time to negative conversion (ANOVA: F = 4.467; *p* = 0.005). Using Bonferroni multiple comparisons, TB patients whose baseline sputum microscopy grade was +1 converted significantly faster than patients with a microscopy grade of +3 on treatment initiation (*p* = 0.009). In addition, when comparing time to negative conversion between TB patients whose sputum microscopy grades ranged from scanty to +1 (5.20 ± 2.36 weeks) and those whose sputum microscopy grades ranged from +2 to +3 (6.70 ± 3.07 weeks), a T-test for independent samples showed that the difference was statistically significant (*p* = 0.0003). For the analysis, data on eight patients were missing.

However, when comparing HIV-positive and HIV-negative patients coinfected with TB with respect to whether they converted negative or remained positive at 4, 8, and 12 weeks, the difference was not statistically significant.

### 3.3. Prolongation of Smear Positivity at the End of 12 Weeks Post-Anti-TB Initiation in Patients Diagnosed with MDR-TB as Compared to Patients Diagnosed with Drug-Susceptible TB

In a bivariate analysis, Table 7 below shows that the proportions of MDR-TB patients with prolonged sputum smear positivity were significantly associated with the baseline smear microscopy grade (*p* = 0.014) and initial CD4 T-cell categories (*p* = 0.010). These differences were not observed among DS-TB patients at the end of 12 weeks post-anti-TB initiation.

In addition, time to microscopy conversion in patients diagnosed with MDR-TB was significantly prolonged as compared to patients diagnosed with DS-TB irrespective of their HIV status using ANOVA (*p* = 0.035) as shown in Table 8. However, these findings were not supported by Bonferroni multiple comparisons (*p* > 0.05) and the Wilcoxon rank sum test (*p* = 0.204). The latter test indicated that there was no significant difference in time to microscopy conversion when comparing survival curves of MDR-TB patients coinfected with HIV (6.78 ± 0.399), MDR-TB patients without HIV infection (7.48 ± 0.663), DS-TB patients coinfected with HIV (5.78 ± 0.350), and DS-TB patients who are HIV-negative (6.036 ± 0.387) (Chi-square = 4.592; *p* = 0.204).

Figure 5 below indicates that the mean time to sputum smear conversion from positive to negative was inversely associated with age among MDR-TB patients. Using a T-test for independent samples involving MDR-TB patients, time to microscopy conversion in patients aged <35 years was significantly prolonged (6.67 ± 3.12 weeks) as compared to patients aged >35 years (5.70 ± 2.63 weeks) (*p* = 0.021).

Finally, at the end of 8 weeks of anti-TB therapy, all MDR-TB patients who had a baseline smear microscopy ranging from scanty to +1 converted negative, while 25% of all MDR-TB patients with smear grade ranging from +2 to +3 remained positive until the end of 12 weeks following treatment initiation (*p* = 0.014).

As depicted in Figure 6 below, all MDR-TB patients who had a baseline smear microscopy from scanty to +1 converted faster (5.20 ± 0.305 weeks; CI: 4.602–5.798) than MDR-TB patients who had a baseline smear microscopy graded from +2 to +3 (7.00 ± 0.272 weeks; CI: 6.466–7.534) using the Wilcoxon rank sum test (Chi-square = 15.543; *p* = 0.0001).

In a multivariate analysis, Cox regression showed that only the baseline sputum smear microscopy grade was independently associated with prolonged smear positivity in MDR-TB patients at the end of 12 weeks following anti-TB initiation (B = −0.405; SE = 0.159; Wald = 6.464; Exp (B) = 0.667; *p* = 0.011) after adjusting for HIV status, CD4+ T-cell count, and age.

## 4. Discussion

In South Africa and other countries with limited resources, despite numerous efforts and progress made, DR-TB remains not only a public health challenge but also an economic challenge because the management of DR-TB cases is extremely costly as compared to drug-sensitive TB [5,16]. This situation is worsening by the presence of a large HIV epidemic in these settings and also by the astronomic rise of cases of MDR-TB and XDR-TB [5,16]. Because regular cultures of patient specimens are not cost-effective in order to monitor patient response to therapy, smear conversion is still effectively used as an indicator in many countries with limited resources [17]. Healthcare workers in settings with limited resources rely on findings from smear microscopy grading to make clinical decisions. The present study was conducted in order to gain insight into delayed time and rate of sputum smear conversion, particularly in the face of dual MDR-TB/XDR-TB and HIV epidemics, as a proxy indicator of the persistence level of TB infectiousness. Factors associated with delays in smear conversion from positive to negative were also explored.

Anecdotal evidence has been suggesting that DR-TB patients, when coinfected with HIV, become more infectious with less response to treatment as compared to DR-TB patients who are HIV-negative. Although previous studies conducted elsewhere have refuted this assumption, it was necessary to test the hypothesis in settings where dual DR-TB and HIV epidemics are prevalent and resources are limited. The underlying assumption was that in resource-limited settings, the use of baseline smear microscopy grading might play a key role in risk stratification before treatment initiation of MDR-TB patients, hence supporting the rational use of basic IPC supplies (i.e., surgical and N95 masks) due to their regular shortage to effectively prevent the transmission of DR-TB.

Our results show that after 8 weeks of appropriate anti-TB treatment, there was no significant difference in the smear microscopy conversion rates of HIV-infected patients as compared to HIV-uninfected patients. Furthermore, it is shown that after 8 weeks of treatment, one patient out of every six treated patients remained potentially infectious, irrespective of the HIV status. This is contrary to the general belief that patients become noninfectious after two weeks of the standard treatment regimen. This finding is in line with other results, which have also shown that the conversion to a negative test, hence the loss of infectiousness of pulmonary tuberculosis patients during therapy, does not occur rapidly in all patients [18,19]. In Tanzania, it was found that after 2 weeks of treatment conversion was significantly higher in HIV-positive (72.8%) than HIV-negative (63.3%) patients [19]. From our results, it is also shown that after 2 weeks of treatment, the conversion rate was significantly higher in HIV-positive (56.6%) than HIV-negative (43.4%) patients. Previous studies have shown that HIV patients do not show delayed sputum conversion [20,21]. On the contrary, it has been argued that HIV patients may have early sputum conversion [21]. A study that was performed in the Karonga district in Malawi showed that the presence of HIV infection was associated with a shorter time to sputum smear conversion [22]. In this study, HIV-negative TB patients were at higher risk for 8-week sputum smear nonconversion than their HIV-positive counterparts. It could be that the specialist-driven treatment-adherence training received by HIV-positive patients also had a positive effect on these patients when they initiated their TB treatment, consequently influencing their sputum conversion.

Moreover, previous research in Tanzania [19] and Uganda [20] has shown that HIV status is not a significant predictor of two-month sputum smear nonconversion. Coinfection with HIV has not been reported to be a significant factor for the smear conversion rate or time to conversion [18,19]. HIV coinfection did not significantly influence the duration of infectiousness during treatment. In this present study, although there was no significant difference in the sputum smear conversion rates of HIV-infected patients versus HIV-uninfected patients, the conversion rate was, however, higher (>90%) among HIV-infected patients whose baseline microscopy grade ranged from scanty to 2+ as compared to patients who had a grade of 3+ at baseline (conversion rate of 76%).

At 8 weeks, 82.5% of all our study participants converted negative. Two separate studies in India reported two widely different conversion rates at the end of 8 weeks: one reported 58% [23], while another reported 84% [24]. It is estimated that more than 20% of smear-positive patients remain infectious after 2 months of treatment [25,26]. Studies performed in many settings have shown that the proportions of sputum smear nonconversion at the end of the intensive phase of TB treatment range from 5% to 32% [27,28].

In this study, acid-fast smear positivity status was also assessed in correlation with CD4+ T-cell counts. Paradoxically, HIV-positive patients with CD4+ T-cell counts < 200 cells/mm^3^ converted faster than those with CD4+ T-cell counts ≥ 200 cells/mm^3^. All HIV-infected MDR-TB patients who had a baseline CD_4_+ T-cell count ≥ 350 cells/mm^3^ at the beginning of the anti-TB treatment had a sputum microscopy conversion from positive to negative after 12 weeks. There is a clear benefit to the individual in starting HIV treatment before reaching a CD4+ T-cell count of 350 cells/mm^3^. In 2012, SA raised the treatment threshold to a CD4+ T-cell count from 200 to 350 cells/mm^3^ for all patients, while pregnant women and anyone with TB are given ART irrespective of the CD4+ T-cell count [29].

Currently, the duration of MDR-TB infectiousness after the initiation of effective treatment is still a subject of discussion. Measures are to be maintained until noninfectiousness has been demonstrated. According to the Centers for Disease Control and Prevention guidelines [30], patients with DS-TB may be considered to be noninfectious when (i) they have completed at least two weeks of standard anti-TB therapy, preferably with direct observation by a TB program-appointed treatment supervisor; (ii) they have demonstrated clinical improvement; and (iii) there is negligible chance of multidrug resistant TB (MDR-TB). However, the World Health Organization’s compendium of indicators for monitoring and evaluating national TB programs states that the majority of new smear-positive pulmonary TB patients should convert to smear-negative after 2 or 3 months of treatment [2,3,6]. In addition, the demonstration of noninfectiousness is best performed by demonstrating culture conversion [2,3,6]. Unfortunately, the time required to report culture results and the availability of resources are major limitations to the use of cultures for the purpose of infection prevention and control. On the other hand, sputum smear microscopy, though less sensitive than cultures according to the literature, can be reported much earlier, but the presence of positive microscopy does not necessarily indicate the presence of living mycobacteria, hence the risk of infectiousness.

For most of our patients who were diagnosed with DS-TB versus MDR-TB, the conversion rate was 91% and 75%, respectively. A study by DeRiemer et al. reported that for managing MDR-TB patients, the TB program can achieve a cure rate of 73% [31]. The present study shows that potential TB infectiousness decreases from the baseline to 56.5%, 82.5%, and 83.0% at the end of 4 weeks, 8 weeks, and 12 weeks, respectively. Our smear conversion rate at the end of 12 weeks (83%) was shown to be higher than the reported 80.0% in Taiwan [32] but lower than the 98.6% found during a study in Tanzania [19] and also lower than the 92% rate reported by Bawri et al. [20].

The present study shows that the age group <35 years was a significant predictor of nonconversion of sputum smears among our smear-positive TB patients after the initial anti-TB treatment. This is contrary to numerous other studies that reported that the elderly were the least likely to have documented sputum conversion [25,33,34,35]. The authors have identified older age as a significant risk factor for sputum smear nonconversion at the end of the intensive treatment phase and also showed that age ≥ 40 years was an independent predictor of nonconversion of sputum smears [25,33,34,35]. Singla et al. observed in a similar study that patients aged over 60 years had an almost six times greater risk of remaining sputum-positive after two months of treatment than patients aged 21–40 years, while patients aged 41–60 years were twice as likely to remain sputum-positive [25]. The reasons for these differences are not clear.

An independent risk factor for nonconversion identified in this present study was a higher smear grade (≥2+) at baseline, and this was irrespective of HIV status. A higher smear grade at baseline has also been identified in other studies as a predictor for nonconversion after 2 months of treatment [4,17,25,26,36]. According to the authors, a heavy initial bacillary load has been documented as an important risk factor for delay in sputum smear conversion at the end of the intensive phase of TB treatment. They argued that delayed treatment onset is associated with a higher bacillary load at diagnosis, which in turn is related to higher sputum smear nonconversion [4,17,25,26,36].

Many other reasons can also explain the nonconversion of sputum smear at the end of 12 weeks of treatment. First of all, nonviable bacteria can remain visible under the microscope. Ideally, culture is the best method to evaluate the viability of *M. tuberculosis* [37]. Unfortunately, this method is not practical in resource-limited settings, such as in developing countries where TB is prevalent. Moreover, a study has shown a good correlation between cultures and sputum acid-fast bacilli smears [22].

Other potential explanations of nonconversion of sputum smears at the end of 12 weeks of treatment are poor supervision of the initial phase therapy, poor treatment adherence by patients, doses of anti-TB drugs below the recommended range, comorbid conditions, drug-resistant *M. tuberculosis* that is not responding to first-line treatment, and heavy initial bacillary load [4,17,25,26,36]. The presence of lung cavitation in a TB patient is a well-recognized risk factor for delayed conversion and treatment failure in TB due to decreased penetration and antibacterial activity of drugs [21,38,39].

The intensive phase completion conversion rate of 83% demonstrated in this study compares well with the expected rate, ranging between 80% and 90%, among MDR-TB patients who are put on appropriate therapy and comply fully with treatment instructions. With full patient compliance and strict adherence to therapeutic instructions, it is anticipated that 80 to 90% of infected patients will respond to therapy and become smear-negative within 2–3 months of treatment [40].

## 5. Conclusions

This study demonstrates that a reasonable number of MDR-TB patients did not convert to a negative smear result at the end of 8–12 weeks and that HIV-seropositive status is not a principal factor in delaying sputum conversion among patients receiving intensive phase tuberculosis treatment. We found that a heavy initial bacillary load is associated with sputum smear nonconversion at the end of the intensive phase of TB treatment. As a result, MDR-TB patients with a heavy initial bacillary load should thus be closely monitored, and healthcare professionals should maintain infection prevention measures until conversion occurs.

The adoption therefore in developing countries of modern rapid methods for identifying and testing the susceptibility of *M. tuberculosis* to at least rifampicin should be encouraged. These will help considerably in the risk stratification of MDR-TB patients, hence improving clinical and infection prevention decision making. In addition, strict monitoring and evaluation of drug adherence especially among MDR-TB patients as well as periodic surveillance of regional drug-resistant status among TB patients are warranted.

Infection prevention and control measures are recommended for all sputum smear-positive patients to minimize the spread of infection. Measures are to be maintained until noninfectiousness has been demonstrated. Patients with a heavy initial bacillary load should thus be closely monitored.

Limitations of the study included the presence of missing data, mainly data on CD4+ T-cell count and HIV plasma viral load results. Further, culture is the best method to evaluate the viability of *M. tuberculosis* and, if performed, could have helped in drawing more accurate conclusions. However, we compared patients with DS-TB versus MDR-TB smear conversion rates and time to conversion and used multivariate Cox regression models to rule out confounders.

## Figures and Tables

**Figure 1 pathogens-12-01133-f001:**
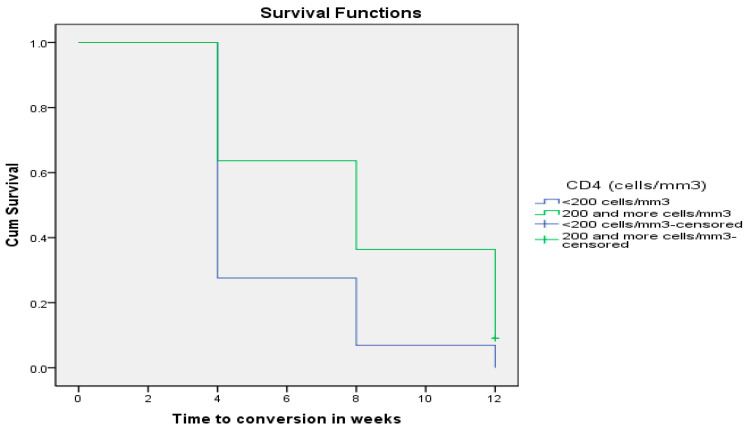
Sputum smear time to conversion (in weeks) by levels of immune suppression (CD4+ T-cells).

**Figure 2 pathogens-12-01133-f002:**
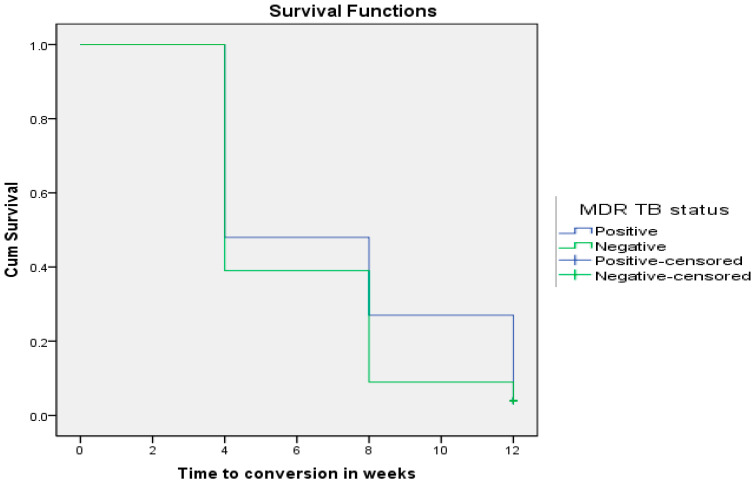
MDR-TB status as a determinant of sputum smear conversion rate in TB patients (*p* = 0.048).

**Figure 3 pathogens-12-01133-f003:**
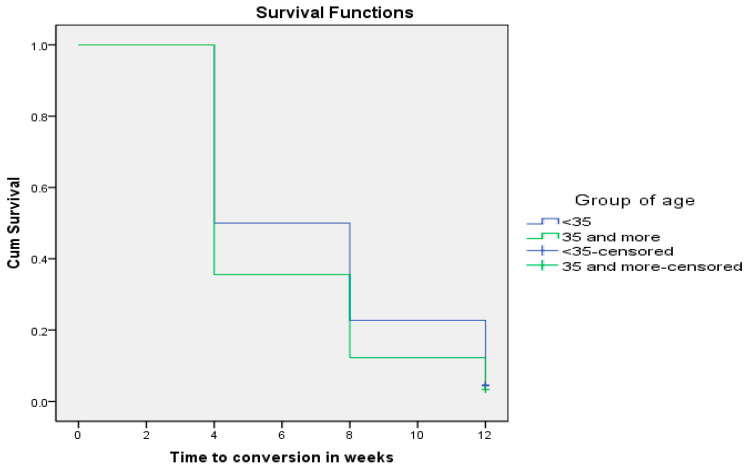
Age as a determinant of sputum smear conversion rate in TB patients (*p* = 0.027).

**Figure 4 pathogens-12-01133-f004:**
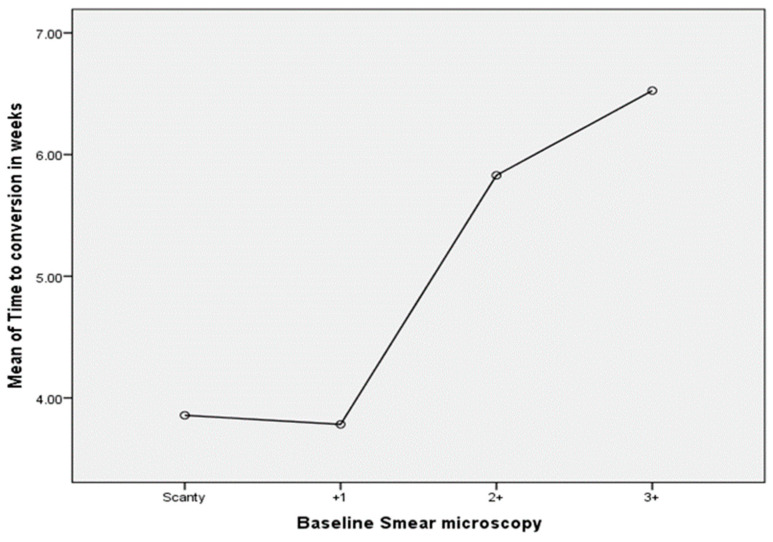
Time to negative conversion by baseline smear grades (in weeks) for all TB patients.

**Figure 5 pathogens-12-01133-f005:**
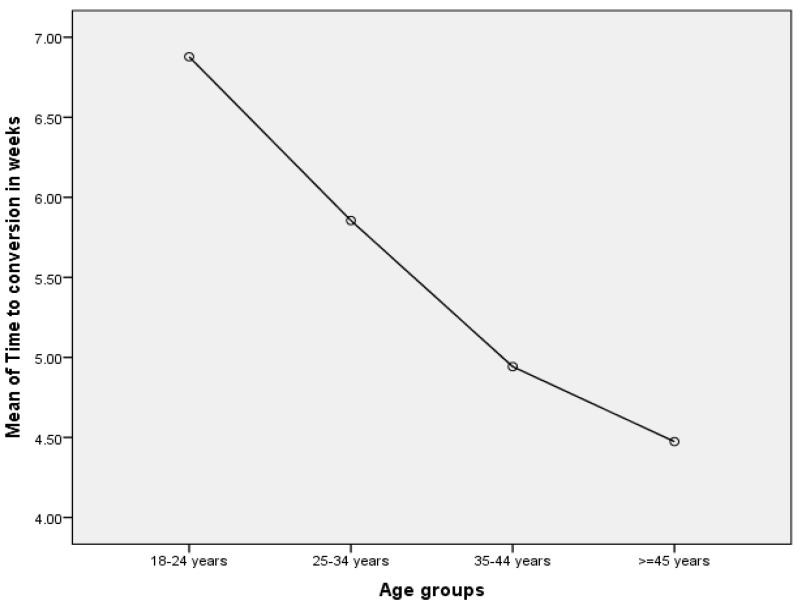
Time to conversion (in weeks) by age groups among MDR-TB patients.

**Figure 6 pathogens-12-01133-f006:**
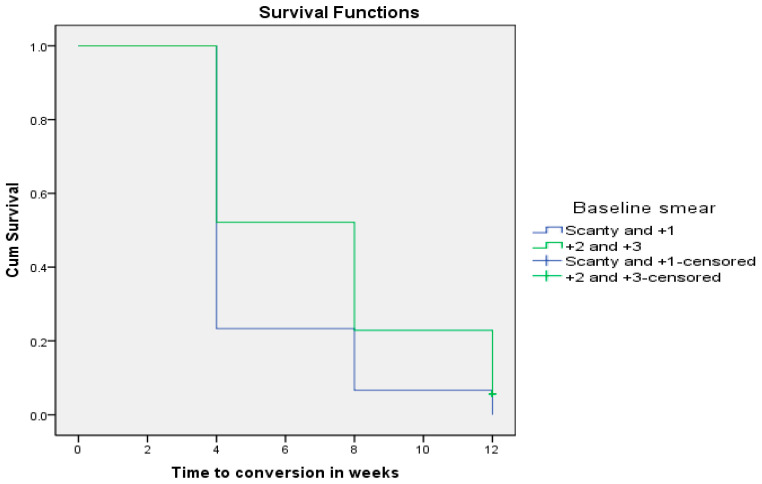
Baseline smear microscopy grade as a determinant of sputum smear conversion rate in MDR-TB patients (*p* = 0.0001).

**Table 1 pathogens-12-01133-t001:** Baseline characteristics of study participants.

Characteristics	Total Number (N)	(%)
Gender		
Male	103	51.5
Female	97	48.5
Age		
<35 years	145	72.5
≥35	55	27.5
HIV status		
HIV (−)	86	43
HIV (+)	114	57
TB susceptibility		
MDR-TB	100	50
DS-TB	100	50

**Table 2 pathogens-12-01133-t002:** Comparison of microscopy smear grades at baseline by HIV status.

Baseline Smear Grade	HIV-Positive	HIV-Negative	Total
Scanty	7	7	14
+1	24	22	46
+2	24	17	41
+3	59	40	99
Total	114	86	200

(Chi-square = 1.029; *p* = 0.794).

**Table 3 pathogens-12-01133-t003:** Proportions of negative sputum smear conversion after 12 weeks of treatment stratified by HIV status.

Variables of Interest	Total	Negative Sputum Smear Conversion after 12 Weeks		*p*-Value
		HIV-Positive	HIV-Negative	
Gender				0.397
Male	103	54/61 (88.5%)	35/42 (83.3%)	
Female	97	43/53 (81.1%)	34/44 (77.3%)	
Age (years)				0.693
≥35	90	48/55 (87.3%)	32/35 (91.4%)	
<35	110	49/59 (83.1%)	37/51 (72.5%)	
Smear microscopy grade				0.563
Scanty	14	7/7 (100%)	7/7 (100%)	
+1	46	22/24 (91.7%)	21/22 (95.5%)	
+2	41	23/24 (95.8%)	14/17 (82.4%)	
+3	99	45/59 (76.3%)	27/40 (67.5%)	
Baseline smear categories				0.503
Scanty and +1	14	7/7 (100%)	7/7 (100%)	
+2 and +3	186	90/107 (84.1%)	62/79 (78.5%)	
TB status				0.0013
MDR-TB	100	54/69 (78.3%)	21/31 (67.7%)	
DS-TB	100	43/45 (95.6%)	48/55 (87.3%)	

**Table 4 pathogens-12-01133-t004:** Linear trend between levels of immune suppression by means of CD4+ T-cell count (cell/mm^3^) and sputum smear conversion status among MDR-TB patients.

CD4+ T-Cell Count (cells/mm^3^) Categories	Total	Sputum Smear Conversion Status		*p*-Value
		Converted Negative	Remained Positive	
≤200	28	26 (52%)	2 (4%)	0.244
200–349	17	10 (20%)	7 (14%)	
350–499	5	5 (10%)	0 (0%)	

(Chi-squared test for Trend statistic = 1.356; *p* = 0.244).

**Table 5 pathogens-12-01133-t005:** Factors associated with prolongation of sputum smear positivity in all TB patients at 12 weeks post-anti-TB initiation.

Variables of Interest	Converted Negative n (%)	*p*-Value
Gender		0.186
Male	89 (86.4)	
Female	77 (79.4)	
Age, years		0.045
≥35	80 (87.9)	
<35	86(78.2)	
Baseline smear microscopy		<0.0001
Scanty	14 (100)	
+1	43 (93.5)	
+2	37 (90.2)	
+3	77 (72.7)	
TB susceptibility		0.003
MDR-TB	75 (75)	
DS-TB	91 (91)	
HIV status		0.365
Positive	97 (85.1)	
Negative	69 (80.2)	

**Table 6 pathogens-12-01133-t006:** Time to negative conversion (in weeks) by baseline sputum smear grades for all patients diagnosed with TB.

Baseline Sputum Smear Grades	n	Mean Time to Negative Conversion (Weeks)	Standard Deviation (SD)	95% Confidence Interval (CI)
Scanty	13	4.92	1.754	3.86–5.98
+1	47	5.28	2.517	4.54–6.02
+2	39	6.15	2.401	5.38–6.93
+3	93	6.92	3.291	6.25–7.60
Total	192	6.23	2.943	5.81–6.65

(ANOVA: F = 4.467; *p* = 0.005).

**Table 7 pathogens-12-01133-t007:** Factors associated with prolongation of smear positivity at 12 weeks post-anti-TB initiation in patients diagnosed with MDR-TB as compared to DS-TB patients.

Variables of Interest	MDR-TB Patients			DS-TB Patients		
	Converted Negative (%)	Remained Positive (%)	*p*-Value	Converted Negative (%)	Remained Positive (%)	*p*-Value
Gender			14 (29.2)			0.266
Male	41/52 (78.8)			48/51 (94.1)	3 (5.9)	
Female	34/48 (70.8)			43/49 (87.8)	6 (12.2)	
Age, years			0.101			0.356
≥35	35/42 (83.3)	7 (16.7)		45/48 (93.8)	3 (6.3)	
<35	40/58 (69.0)	18 (31.0)		46/52 (88.5)	6 (11.5)	
Baseline smear			0.014			0.452
Scanty	8/8 (100)	0 (0)		6/6 (100)	0 (0)	
+1	15/17 (88.2)	2/17 (11.8)		28/29 (96.6)	1/29 (3.4)	
+2	13/14 (92.9)	1/14 (7.1)		24/17 (88.9)	3/27 (11.1)	
+3	39/61 (63.9)	22/61 (36.1)		33/38 (86.8)	5/38 (13.2)	
Baseline smear			0.089			0.427
Scanty and +1	8/8 (100)	0 (0)		6/6 (100)	0 (0)	
+2 and +3	67/92 (72.8)	25/92 (27.2)		85/94 (90.4)	9/94 (9.6)	
CD4+ T-cell groups			0.010	-	-	-
<200 cells/µL	26 (92.9)	2 (7.1)				
≥200 cells/µL	49 (68.1)	23 (31.9)				
HIV status			0.261			0.150
Positive	54/69 (78.3)	15 (21.7)		43/45 (95.6)	2/45 (4.4)	
Negative	21/31 (67.7)	10 (32.3)		48/55 (87.3)	7/55 (12.7)	

**Table 8 pathogens-12-01133-t008:** Time to microscopy conversion (in weeks) by HIV status in MDR-TB patients versus DS-TB patients.

HIV and MDR-TB Status	n	Mean	Standard Deviation	95% Confidence Interval (CI)
HIV (−) and MDR (−)	52	5.69	2.548	4.98–6.40
HIV (−) and MDR (+)	30	7.33	3.651	5.97–8.70
HIV (+) and MDR (−)	44	5.64	2.168	4.98–6.30
HIV (+) and MDR (+)	66	6.55	3.187	5.76–7.33
Total	192	6.23	2.943	5.81–6.65

(ANOVA: F = 2.920; *p* = 0.035).

## Data Availability

Data and materials pertaining to this study are freely available and can be accessed at Figshare: https://figshare.com/s/92ea41295cbdd81b682f. Last accessed on 5 September 2023.

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
