# Peer review of "Prolongation of Acid-Fast Bacilli Sputum Smear Positivity in Patients with Multidrug-Resistant Pulmonary Tuberculosis"

_pathogens, 2023, doi:10.3390/pathogens12091133_

Round 1

Reviewer 1 Report

I reviewed Sidwell Mvo et al paper entitled Prolongation of acid-fast bacilli sputum smear positivity in patients with multidrug-resistant pulmonary tuberculosis and overall, the paper is well organized with interesting findings. My questions to the authors are the following: 

1.     How were the 200 patients selected for screening? 

2.     Explain how the sample size was determined. 

3.     Do patients checked for other diseases which reduce CD4+ count? If not, the estimation of CD4+ count due to the immunosuppression may not reflect the actual percentage. Explanation is required here.

4.     Figure legends could be expanded and explained more. 

Reviewer 2 Report

Wao et al. have asked an essential question in the field of clinical tuberculosis. The authors have studied the effect and association of age, HIV positivity, smear grade, and CD4 T-cell count (extent of immunosuppression) with the sputum conversion. They have shown that multidrug-resistant tuberculosis patients tend to have prolonged sputum positivity compared to other groups in a cohort of 200 patients comprising MDR and DS-TB patients with and without HIV co-infections. Both the study and the observations made are straightforward. However, the representation of the data is highly convoluted and needs improvement.

Overall, I recommend the publication of the manuscript following the resolution of the concerns raised.

Major comments:

Line 192-199, The authors should provide the specifics of the patients also as an additional table

The authors have described the factor associated with the sputum positivity rate in MDR-TB patients. Did the author also perform the same analysis for Drug-sensitive TB patients?

Table 3: depicts the total number of patients per column with the association but not the total number per group. Please follow the same format as Table 2 for a consistent data representation.

The graphs in Figures 1, 2, and 3 are complicated to interpret. The figure legends are unclear too. What do authors mean by censored vs uncensored? Are there four or only two groups, as only two lines are visible on the graph? Please clarify

Table 5: where is the data on the eight patients as the total number of patients seems to be only 192

The resolution of the graphs needs further improvement. Please make sure to use at least a 300 dpi resolution.

The language can be simplified.
